# Digital Twin Virtual Welding Approach of Robotic Friction Stir Welding Based on Co-Simulation of FEA Model and Robotic Model

**DOI:** 10.3390/s24031001

**Published:** 2024-02-04

**Authors:** Shujun Chen, Guanchen Zong, Cunfeng Kang, Xiaoqing Jiang

**Affiliations:** College of Mechanical and Energy Engineering, Beijing University of Technology, Beijing 100124, China; sjchen@bjut.edu.cn (S.C.); zongguanchen@emails.bjut.edu.cn (G.Z.); xjiang@bjut.edu.cn (X.J.)

**Keywords:** RFSW, robot stiffness, CEL, deviation measurement, welding process simulation

## Abstract

Robotic friction stir welding has become an important research direction in friction stir welding technology. However, the low stiffness of serial industrial robots leads to substantial, difficult-to-measure end-effector deviations under the welding forces during the friction stir welding process, impacting the welding quality. To more effectively measure the deviations in the end-effector, this study introduces a digital twin model based on the five-dimensional digital twin theory. The model obtains the current data of the robot and six-axis force sensor and calculates the real-time end deviations using the robot model. Based on this, a virtual welding model was realized by integrating the FEA model with the digital twin model using a co-simulation approach. This model achieves pre-process simulation by iteratively cycling through the simulated force from the FEA model and the end displacement from the robot model. The virtual welding model effectively predicts the welding outcomes with a mere 6.9% error in lateral deviation compared to actual welding, demonstrating its potential in optimizing welding parameters and enhancing accuracy and quality. Employing digital twin models to monitor, simulate, and optimize the welding process can reduce risks, save costs, and improve efficiency, providing new perspectives for optimizing robotic friction stir welding processes.

## 1. Introduction

As environmental concerns intensify, reducing carbon emissions and achieving sustainable development have become focal issues. Exploring eco-friendly novel composite materials [1] and developing low-emission manufacturing methods, such as 3D printing [2], are current research hotspots. Traditional welding techniques are marred by high energy consumption and significant pollution, but friction stir welding (FSW) technology has marked a transformative change. FSW technology was invented in 1991. As a solid-state joining process, FSW possesses a low welding temperature and minimal residual stress, and it has been widely used in fields such as aviation, automotive, railway, maritime, and more [3,4]. Especially in the field of dissimilar metal welding, such as aluminum–magnesium, aluminum–copper, carbon steel–stainless steel, etc., FSW technology demonstrates irreplaceable advantages [5,6]. The development of FSW technology has progressed rapidly with significant contributions from many scholars. Rahmi et al. [7] proposed an improved FSW technique named friction stir vibration welding (FSVW). This method enhances joint quality and performance by applying vibration to the weldments. Abbasi et al. [8] demonstrated the advantages of FSVW over traditional FSW in producing tailor-welded blanks (TWBs). Although FSW technology has many benefits compared to conventional welding techniques, its higher welding force limits its application in more fields. Additionally, it requires higher demands on welding equipment.

At present, FSW equipment is mainly developed based on NC machines, and it is difficult to achieve complex welding work [9]. To enhance the performance of FSW equipment and expand its application fields, a 5-DOF gantry robot with a slide rail and rotary table was developed for FSW [10]. Its design and kinematic model were validated through simulations of melon petal welding conditions. Though there are many advantages of dedicated FSW equipment, the cost of its research, development, and manufacturing is excessively high [11]. Therefore, combining FSW equipment with universal heavy payload serial industrial robots can achieve lower development costs, a larger working space, and more flexible work tasks [12]. However, for several years, many problems prevented the application of robotic friction stir welding, especially the problem of excessive end-effector deviation caused by the low stiffness of serial industrial robots [13]. To solve this problem, some researchers proposed a method to construct a hybrid stiffness index, optimizing the structural dimensions and trajectory of the robot, which achieved effective results [11]. However, stiffness optimization alone does not eliminate end-effector deviation in welding. The accurate detection and compensation of end-effector displacement are crucial for enhancing welding quality. Some researchers employ online detection methods for the direct or indirect detection and compensation of end-effector deviation. Li et al. [14] presented a multiparameter sensing method for the five-axis RFSW based on laser circular scanning and verified its feasibility in deviation detection. Soron et al. [15] established a robotic friction stir welding system based on a force control algorithm on the robot platform and experimentally verified its feasibility. However, the online detection and error compensation often exhibit latency and are less effective when performing high-speed or complex curve welding. To advance compensate for the end-effector deviation in the robot, a deflection model-based feedforward compensation technique and an offline path planning using Bézier curves were proposed to solve the position and orientation deviations in the robot end-effector during welding [16]. Well, most such studies assume that the stirring tool is subjected to a fixed force value during the welding process without considering the variations in the force conditions during the welding process.

With the advent of Industry 4.0, various cutting-edge technologies such as cyber-physical systems, digital twin, IoT, robots, big data, and cloud computing have emerged. However, the true potential of Industry 4.0 lies in the collaborative and enhanced functionality achieved through the interconnection of these technologies. Among these combinations, integrating digital twin into robotics is particularly novel but holds unparalleled possibilities [17]. With the development of digital twin technology and the application of computer science, artificial intelligence, and other technologies in the field of digital twin, new approaches have been provided to address the challenges encountered by robots in industrial production [18]. Through the digital twin approach, a solution employing Knowledge Graph and Function Block methods was proposed to address the low stiffness of robotic machining for large-scale components (LSCs) [19]. Chen et al. [20] introduced a digital twin modeling approach for FSW using sensor-based numerical simulation, achieving the synchronization of the simulation and physical process with a high temporal precision for effective cyber-physical integration. Sigl et al. [21] proposed a new welding temperature feedback system for the FSW process, using a digital twin and an adaptive controller. It demonstrates improved weld quality through experimental validation. Digital twin technology is also applied in the design of FSW tools [22]. The development of digital twin technology and computer simulation techniques has given rise to virtual machining technology. It is a computer-based simulation approach that models and analyzes the machining process before actual machining occurs. It helps optimize tool paths, reduce material waste, and improve product quality. Kang et al. [23] established a virtual machining model using an optimization method to calculate the cutting forces during milling. Jauhari et al. [24] built a digital twin virtual machining model for milling chatter detection. The model achieved an average classification accuracy of 94.04% and offered a novel method for the manufacturing industry to improve data monitoring conditions in machining.

Digital twin technology bridges the real and digital worlds, enabling observation and interaction with the physical world digitally. This offers new perspectives for RFSW. Digitizing the RFSW process enables overcoming the difficulty in directly measuring robot end deviations, allowing for real-time observation and measurement through virtual entities. Additionally, virtual machining technology can overcome the limitation of dynamically reflecting the welding process, allowing for the accurate prediction of welding outcomes and robot end deviations. This study proposed a digital model for RFSW. The model utilized a co-simulation approach integrating the finite element analysis model and the robot kinematic and stiffness model to reproduce the welding process digitally. Before fabrication, it employed a virtual welding process to predict the welding results and deviations under specific parameters. This model can also compensate for welding errors and optimize parameters before welding. The rest of this study is organized as follows. Section 2 establishes a five-dimensional digital twin model detailing Physical and Virtual Entity modules and creates a virtual welding model through co-simulation. Section 3 involves calibrating digital twin data with actual data from the Physical Entity. In Section 4, the analysis of the virtual welding results under different parameters is conducted and compared with the actual welding outcomes. Section 5 summarizes the study.

## 2. Digital Twin Virtual Welding Model of Robotic Friction Stir Welding

To achieve virtual welding for robotic friction stir welding, it is imperative to construct a digital twin model of robotic friction stir welding, which establishes a mapping relationship between the virtual space and the physical space and creates bidirectional data connectivity.

Digital twin refers to a virtual representation of a physical process or product capable of boosting efficiency and cutting costs within manufacturing processes [25]. The concept of “Twin” was first employed during NASA’s Apollo project in manufacturing. It involved creating identical spacecraft, one on Earth mirroring the mission craft. The “Twin” aided training, simulated verification, and predicted the mission craft’s status, assisting astronauts with accurate operations [26]. A digital twin was expanded on the “Twin” idea from the Apollo project into the virtual realm. It formed a virtual product mirroring the physical item in looks and essence using digital techniques. This links the virtual and physical realms, enabling the exchange of data and information [26].

This study established a five-dimensional digital twin model to make the robotic friction stir virtual welding simulation model more standardized, operable, and portable. The five-dimensional digital twin framework suggests that a digital twin model includes the following dimensions: the Physical Entity (PE), the Virtual Entity (VE), the Services module (Ss), the digital twin data (DD), and connections (CNs) [27]. Each module is defined in Figure 1.

The Physical Entity module consists of a friction stir welding robot and the actual friction stir welding conditions. The Virtual Entity module contains the simulation model of the Physical Entity. Digital twin data include the actual data from the Physical Entity, such as the robot end-effector’s forces and the joint’s angles, and the Virtual Entity’s simulated data, such as the simulated forces and the deviations in the robot end-effector. The Service module is integrated with the virtual welding model of robotic stir welding, which contains the finite element analysis model, the robot kinematic model, and the robot stiffness model. The connections include physical and logical connections, which can ensure the communication of each module and the operation of the whole model.

### 2.1. Physical Entity Module

The Physical Entity encompasses the tangible elements within the digital twin framework, responsible for generating functional outputs within the digital twin’s operational processes. Additionally, the PE is tasked with acquiring the original environmental and operational data. This module includes the friction stir welding robot, and the environments of friction stir welding. The robotic part of friction stir welding consists of a robot, a six-axis force sensor, an electric spindle, and a stirring tool. The robot is a heavy-duty industrial robot model ZK-500, produced by Zhen Kang Machinery Co., Ltd., Nantong, China, characterized by a weight of approximately 2400 kg, a maximum operational range of 3000 mm, and a rated load capacity of 500 kg. It incorporates a six-axis force sensor from SRI, model M4347D1, capable of measuring forces up to 15 KN and torques up to 6 KNm. The electric spindle operates at a rated speed of 5000 rpm, with a rated torque of 55 Nm, a weight of 150 kg, and can withstand a maximum load of 5000 N. The appearance and dimensional parameters of the ZK-500 robot are shown in Figure 2.

The environment of friction stir welding mainly includes the types of base material for welding, welding parameters, welding path, and environmental temperature. This study took the example of the flat linear welding of a 5 mm thick 6061 aluminum alloy plate using a conical stirring tool with a rotation speed of 1500 rpm, a welding speed of 3 mm/s, and a tilt angle of 2 degrees. The environmental temperature is about 24 °C. The weld seam can achieve good surface quality through experimental verification and exhibit distinct, measurable lateral and advance resistance forces. The result is shown in Figure 3.

The length of the pin of the conical stirring tool is 4.8 mm. In the plunging stage, the actual plunging depth was suitable for the length of the stirring tool. However, the motion command of the robot end-effector was about 8 mm. The target trajectory of the weld seam is shown in the green dashed line in Figure 3b. In the welding stage, the robot end-effector experienced a displacement of about 5 mm. The actual trajectory is the yellow dashed line. Through the analysis of force distribution during the friction stir welding process, the stirring tool experienced axial force ($F_{d}$), lateral force ($F_{l}$), and forward resistance force ($F_{r}$) during the welding process can be inferred, as shown in Figure 3c. The variation in the welding force during the whole process is shown in Figure 3d. The axial force was the largest one, about 5000 N. The lateral and resistance forces were about 1000 N at the welding stage. The displacement of the robot end-effector resulted from the huge welding force. It is easier to measure the welding force than the displacement. The displacement of the robot end-effector can be calculated by measuring the force on the robot end-effector during the welding process.

### 2.2. Virtual Entity Module

The end-effector displacement of a serial robot changes with its pose under the same force conditions. To accurately calculate the displacement, it is necessary to establish kinematic and stiffness models of the robot. The result can be seen in the Virtual Entity module.

The Virtual Entity is a real-time model that encompasses geometric and physical models of the Physical Entity’s elements, including its structure, function, and environment. Operators can view the status of the system via the Virtual Entity and transmit instructions to regulate the state of the PE module. A typical VE module can replicate the actual production process in a digital space based on various data outputs by PE, such as encoder data, current data, position data, and so on. However, in robotic friction stir welding, the significant welding force and the low stiffness of the serially connected robots resulted in deviations from the expected trajectory at the end-effector position. Due to the semi-closed-loop control characteristic of industrial robots, the control system cannot directly detect the displacement of the robot end-effector. Therefore, there is a certain deviation between the displayed end-effector position in the VE module and the actual position. To correctly display the position of the robot end-effector within the VE module, this study integrated the robot kinematic and stiffness models with the simulation model. Monitoring data from the six-axis force sensor calculated the displacement of the robot end-effector under the influence of the welding forces and compensated for the robot’s displayed position. The following study involves a detailed study of the robot kinematic and stiffness models.

#### 2.2.1. Kinematic Model of Robot

Establishing an accurate kinematic model forms the foundation of research in robotics. The Denavit–Hartenberg (DH) method is the most commonly used approach for establishing a robot kinematic model. Move each joint of the ZK-500 robot to its initial position and establish a Modified DH (MDH) model, as illustrated in Figure 4 [28].

Based on the MDH model and the dimensional parameters of the ZK-500 robot in Figure 2b, the DH parameters can be obtained in Table 1.

The homogeneous transformation matrix for adjacent link coordinate systems in the MDH model is defined as Equation (1).
(1)$$\begin{array}{c} {\;}_{i}^{i-1}T=R_{x}\left(\alpha_{i-1}\right)T_{x}\left(a_{i-1}\right)R_{z}\left(\theta_{i}\right)T_{z}\left(d_{i}\right)=\left[\begin{array}{cccc} c\theta_{i} & -s\theta_{i} & 0 & a_{i-1}\\ s\theta_{i}c\alpha_{i-1} & c\theta_{i}c\alpha_{i-1} & -s\alpha_{i-1} & -s\alpha_{i-1}d_{i}\\ s\theta_{i}s\alpha_{i-1} & c\theta_{i}s\alpha_{i-1} & c\alpha_{i-1} & c\alpha_{i-1}d_{i}\\ 0 & 0 & 0 & 1 \end{array}\right] \end{array}$$
where $R_{x}$ is the coordinate system rotating around the *x*-axis; $T_{x}$ is the coordinate system translating along the *x*-axis; $R_{z}$ is the coordinate system rotating around the *z*-axis; $T_{z}$ is the coordinate system translating along the *z*-axis; c represents cos; and s represents sin.

According to the definition, the homogeneous transformation matrices between each link can be obtained. Continuously multiplying these matrices as Equation (2) yields the homogeneous transformation matrix from the robot base coordinate system to its end-effector.
(2)T60=T10 21T 32T 43T 54T 65T=nxoxaxpxnyoyaypynzozazpz0001

Based on the robot homogeneous transformation matrix, both the calculations of the forward kinematics and inverse kinematics for the robot can be performed.

It is necessary to establish the robot Jacobian matrix to further investigate the relationship between the robot end-effector and the velocities and forces at each joint. The Jacobian matrix is a linear mapping that relates the operational space velocities of the robot to the joint space velocities. Additionally, it can depict the transmission of forces between the two spaces. According to the cross-product method, the Jacobian matrix of rotary joint i can be written as Equation (3):(3)$$\begin{array}{c} J_{i}=\left[\begin{array}{c} z_{i}\times r_{i}\\ z_{i} \end{array}\right] \end{array}$$
where $z_{i}$ is the unit vector along the joint axis and $r_{i}$ is the radius vector from the center of the joint axis to the endpoint.

The Jacobian matrix of the robot as a whole is $\begin{array}{c} J=\left[\begin{array}{cccc} J_{1} & J_{2} & \ldots & J_{6} \end{array}\right] \end{array}$.

#### 2.2.2. Stiffness Model of Robot

When a robot operates, the end-effector experiences deviations owing to external forces. These deviations in the end-effector resulted from deformations in both the joints and the links. Typically, the stiffness of the robot links was much higher than that of the joints. Therefore, when investigating stiffness-related issues in robots, the deformations of the links are often disregarded [29].

Robot joints are composed of motors and reducers, making the structure relatively complex. When studying the overall stiffness model of a robot, it can be simplified, typically represented using a linear torsion spring to characterize the mechanical structure of the robot joints [11]. Therefore, the joint stiffness model of a six-axis robot can be represented by a diagonal matrix shown in Equation (4).
(4)$$\begin{array}{c} K_{\theta }=diag\left(\left[K_{1},K_{2},K_{3},K_{4},K_{5},K_{6}\right]\right) \end{array}$$

From the statics of the robot, the relationship between the joint torque of the robot and the forces at the end-effector can be derived as Equation (5).
(5)$$\begin{array}{c} \tau =J^{T}F \end{array}$$
where $J^{T}$ is the transpose of the Jacobian matrix and F is the generalized force exerted on the robot end-effector.

When subjected to a torque, the deflection of the robot joints is written as Equation (6):(6)$$\begin{array}{c} dq=K_{\theta }^{-1}\cdot \tau \end{array}$$

Based on the relationship between the robot joint’s deflection and the end-effector’s displacement, the formula for calculating the deviation under the influence of external forces at the robot end-effector is displayed in Equation (7).
(7)$$\begin{array}{c} \Delta X=J\cdot d_{q}=JK_{\theta }^{-1}J^{T}F \end{array}$$

After establishing the robot stiffness model, the next step is to identify the stiffness parameters of the robot joints through experiments. The process for the identification experiment of the robot joint stiffness is as follows:

Firstly, connect the robot end-effector to the workbench using a steel cable through a six-axis force sensor, as shown in Figure 5a. Next, operate the robot in a specific pose where the steel cable is tightened, as in Figure 5b, and record the robot joint angles, end-effector orientation, and force data. Then, the robot will be operated to move its end-effector a short distance in the direction of the steel cable, and the robot’s joint angles, end-effector orientation, and force data after moving will be recorded. After that, calculate the changes in the joint angles and end-effector poses before and after the robot movement. Due to the robot end-effector being securely fixed to the workbench using the steel cable, it can be approximately assumed that the robot end-effector coordinate position has not changed. The changes in the joint angles and end-effector orientation before and after the robot movement can be considered as the pose changes induced by the tension in the steel cables acting on the robot. Finally, calculate the Jacobian matrix for the robot in its current pose, and then put the calculated robot pose change and the force data of the end-effector into the formula of Equation (7) to obtain a set of robot joint stiffness data.

This experiment collected about 50 sensor data points over approximately 5 s after the robot was stabilized to reduce sensor measurement errors. The average of these data points was then calculated to obtain the current force data at the robot’s end. To minimize the impact of the robot’s posture and slight displacements at the robot’s end on the experimental results, this experiment selected five different poses within the robot’s common working range and conducted multiple sets of experiments with varying loading forces for each of these poses as Figure 5b–f. The robot joint angles for different poses are shown in Table 2.

To simplify the calculation process, the robot stiffness matrix can be expressed in the form of a compliance matrix: $C_{\theta }=K_{\theta }^{-1}$

Then, the formula of Equation (7) can be written in Equation (8):(8)$$\begin{array}{c} \Delta X=\left[\begin{array}{c} \sum_{j=1}^{6}\left(C_{j}\;J_{1j}\;\sum_{i=1}^{6}J_{ij}F_{i}\;\right)\\ \ldots \\ \ldots \\ \ldots \\ \ldots \\ \sum_{j=1}^{6}\left(C_{j}\;J_{6j}\;\sum_{i=1}^{6}J_{ij}F_{i}\;\right) \end{array}\right] \end{array}$$

The discriminant $C_{\theta }$ can be separated, and Equation (8) can be written in the form of a system of linear equations as $\begin{array}{c} \Delta X=LC_{\theta } \end{array}$.

The coefficient matrix is as Equation (9):(9)$$\begin{array}{c} L=\left[\begin{array}{cccc} J_{11}\sum_{j=1}^{6}J_{j1}f_{j} & J_{12}\sum_{j=1}^{6}J_{j2}f_{j} & \ldots & J_{16}\sum_{j=1}^{6}J_{j6}f_{j}\\ J_{21}\sum_{j=1}^{6}J_{j1}f_{j} & J_{21}\sum_{j=1}^{6}J_{j2}f_{j} & \ldots & J_{26}\sum_{j=1}^{6}J_{j6}f_{j}\\ \ldots & \ldots & \ddots & \ldots \\ J_{61}\sum_{j=1}^{6}J_{j1}f_{j} & J_{62}\sum_{j=1}^{6}J_{j2}f_{j} & \ldots & J_{66}\sum_{j=1}^{6}J_{j6}f_{j} \end{array}\right] \end{array}$$
where $J_{ij}$ is the Jacobian matrix, row i, column j, and $f_{j}$ is the jth element of the generalized force vector F.

Next, utilize the least squares method to solve the overdetermined system of equations as Equation (10). This involves substituting the robot end-effector forces, and end-effector pose changes into the identification equation to find the parameter ${\tilde{C}}_{\theta }$ that minimizes the error e.
(10)$$\begin{array}{c} \min e\left({\tilde{C}}_{\theta }\right)=\frac{1}{2}\left.\left|\left.|LC_{\theta }-X\right.\right|\right.{|}_{2}^{2} \end{array}$$

The solution yields the robot joint compliance vector as the least squares solution to the system of equations as Equation (11).
(11)$$\begin{array}{c} {\tilde{C}}_{\theta }={\left(L^{T}L\right)}^{-1}L^{T}X \end{array}$$

After obtaining the compliance vector, taking the reciprocal of each of its elements yields the robot joint stiffness vector. The joint stiffness of the robot is shown in Table 3.

Once the determined joint stiffness values are introduced into Equation (7), it becomes feasible to compute the end-effector displacement based on the robot’s applied end-effector forces and compensate for the displayed deviation in the VE. The Virtual Entity module was developed using the Visual Studio 2015 Integrated Development Environment and was created with the Visualization Toolkit (VTK), version 8.2.0. Due to sensor and equipment limitations, device signal fluctuations and external interferences often influence single data collection. To achieve accurate measurements, data must be averaged over a period, compromising the Virtual Entity’s real-time response to equipment dynamic changes. Considering the gradual changes in robot posture and end force in FSW processes, a 0.5 s collection window has been established to balance real-time responsiveness with accuracy in calculating robot end deviations.

### 2.3. Services Module with Co-Simulation Virtual Welding Model

It is necessary to establish a virtual welding model to predict the deviation in the robot end-effector position through virtual welding. The Ss module encompasses user-oriented business and internal functional services to bolster the digital twin’s functionality [27]. Integrate the virtual welding model within the Services module, ensuring its autonomy from the Virtual Entity. This configuration enables the seamless data exchange between physical entities and virtual counterparts while facilitating independent execution to attain virtual welding outcomes before actual operation.

With advancements in finite element simulation technology and the increasing computational capabilities of computers, finite element methods have become a more accurate means to simulate complex physical processes. By integrating a finite element analysis model of friction stir welding with a digital twin model of a robot, a collaborative simulation approach based on a multi-model that combines the finite element analysis model, robot kinematic model, and robot stiffness model can be employed to achieve a virtual welding process for robotic friction stir welding. This approach allows for predicting forces acting on the stirring tool, deviations in the robot end-effector, and the weld seam quality during the friction stir welding process.

#### 2.3.1. Finite Element Analysis Model of Friction Stir Welding

Finite element analysis (FEA) is a numerical technique used to analyze the response of structures and materials to external factors such as forces, heat input, and other physical effects. The friction stir welding process is a typical force–heat coupling process involving temperature changes, the interaction of forces, material deformation, and flow. Common finite element modeling methods for friction stir welding include the heat source model, methods based on Computational Solid Mechanics (CSM), and Computational Fluid Dynamics (CFD) [30]. In the friction stir welding process, the highest temperature is lower than the melting temperature of the base material [31], and there is no solid-to-liquid phase transformation in the base material. The Computational Solid Mechanics approach is theoretically the most suitable for practical applications.

Computational Solid Mechanics is a discipline that employs numerical methods and computational technology to study the mechanical behavior of solid materials and structures. It is a significant branch of computational mechanics, focusing on the mechanical issues of solid materials under the influence of external loads, such as deformation, stress distribution, and failure. The commonly used computational solid mechanics methods in finite element simulations of friction stir welding include the Lagrangian method, the Arbitrary Lagrangian–Eulerian (ALE) method, the Coupled Eulerian–Lagrangian (CEL) method, and the particle method [30]. The CEL method employs a fixed grid while handling material deformation by allowing materials to flow within the grid. Compared to other methods, the CEL method does not lead to grid distortion, making it more suitable for describing significant deformation problems such as friction stir welding. The process of establishing an FEA model for friction stir welding using the CEL method is as follows. The software used for the operation is Abaqus CAE 2021 [32].

Firstly, create a three-dimensional model based on the actual size of the weldment and the stirring tool. The weldment consists of two plates with a dimension of 150 mm × 150 mm × 5 mm to be joined together. Use these dimensions to establish a Euler domain with 300 mm × 150 mm × 13 mm as the space for holding the base material, as shown in Figure 6. Divide the Euler domain into the lower 5 mm thick part for the base material section and the upper 8 mm thick part for the vacuum section, as shown in Figure 6a. During the computational process of the CEL model, materials can freely flow within the Euler domain. The stirring tool model shown in Figure 6b is based on the actual stirring tool, which has a shoulder with a diameter of 15 mm and a stirring pin with a length of 4.8 mm in the form of a conical stirring tool.

In the second step, assemble the weldment model with the stirring tool model so that the stirring tool model is inclined at a 2° angle along the feed direction, as shown in Figure 6c.

The third step is to set the material properties of the base material and the stirring tool. The welding material used in this experiment is AA6061-T6 aluminum alloy, and the material for the stirring tool is H13 tool steel. The stirring tool is made by CFSWT, Beijing, China. The material properties are temperature-dependent to make the results more accurate. The material properties are shown in Table 4 and Table 5 [31,33,34,35].

In the actual welding process, the base material undergoes almost no deformation and can be regarded as a rigid body. The base material is softened under the heat generated by friction with the stirring tool and undergoes deformation and flow under the influence of the welding force. When researching issues related to the FEA simulation of friction stir welding, determining the material constitutive model is essential to ensure accurate simulation results. This study employs the JC model as the computational model for the plastic deformation of materials.

The JC model, short for the Johnson–Cook constitutive model, is an empirical constitutive model commonly used to describe the plastic behavior of metallic materials by varying the temperatures and strain rates. The model can be written as Equation (12) [36,37]
(12)$$\begin{array}{c} \sigma =\left(A+B\varepsilon^{n}\right)\left(1+C\ln\frac{\dot{\varepsilon }}{\dot{\varepsilon_{0}}}\right)\left(1-{\left(\frac{T-T_{room}}{T_{melt}-T_{room}}\right)}^{m}\right) \end{array}$$
where σ is the flow stress; A and B are material constants obtained through experiments; C is the strain rate sensitivity; n is the strain hardening coefficient; m is the thermal softening coefficient; ε is the effective plastic strain; $\dot{\varepsilon }$ is the effective plastic strain rate; $\dot{\varepsilon_{0}}$ is the normalizing strain rate; T is the current temperature; $T_{room}$ is the ambient temperature; and $T_{melt}$ is the melting temperature. The constants of the Johnson–Cook constitutive model for AA6061-T6 are A: 324 MPa, B: 114 MPa, C: 0.002, n: 0.42, m: 1.34, $T_{room}$: 24 °C, $T_{melt}$: 583 °C, and $\dot{\varepsilon_{0}}$: 0.02.

The fourth step involves setting up the analysis steps and the boundary conditions, including the rotation speed of the stirring tool, the feeding speed, and the feeding direction. The final step is to mesh the model of the weldment and the stirring tool. The element used for the weldment is an 8-node thermally coupled linear Eulerian brick, type EC3D8RT; the mesh used for the stirring tool is a 10-node modified thermally coupled second-order tetrahedron, type C3D10MT, with hourglass control. The minimum element is located in the weldment welding area, with dimensions of 0.5 × 0.5 × 0.5 mm.

#### 2.3.2. Efficiency Optimization by Mass Scaling

Efficiency is an essential consideration in engineering applications. As a numerical computational method, finite element analysis often requires significant computational time to obtain accurate results. The solution process for the CEL model employed an explicit analysis method, which, although advantageous in dealing with complex contact and transient analysis problems, also resulted in a notably slow solving process due to the substantial computational load [38].

In explicit analysis methods, the number of increments for each analysis step is determined as n=t/Δt [39].

Where t is the analysis step time; Δt is the stable time increment; and n is the number of increments.

The time increment in the explicit analysis is conditionally stable, and the stability of the operator is determined by the system’s characteristics as Equation (13).
(13)$$\begin{array}{c} \Delta t\leq \frac{2}{\omega_{max}}=f\times \left[\frac{l_{min}}{c}\right] \end{array}$$
where $\omega_{max}$ is the highest frequency or largest eigenvalue of the system; f is a factor to improve the stability; $l_{min}$ is the smallest element dimension in the model; and c is the wave speed.

The wave speed can be determined as $c=\sqrt{\lambda +2\mu /\rho }$

Where λ and μ are Lame’s constant, and ρ is the density of the material.

It can be inferred as Equation (14):(14)$$\begin{array}{c} \Delta t\leq f\times \left[l_{min}\right]\times \sqrt{\frac{\rho }{\lambda +2\mu }} \end{array}$$

From this, it can be deduced that increasing the material density can increase the stable time increment. It can be concluded that increasing the stable time increment (Δt) can reduce the number of increments (n), thereby reducing the computational workload of the computer and speeding up the solving process. Replace the density in the material properties with a fictitious density as $\begin{array}{c} \rho^{*}=k_{m}\rho \end{array}$, and the fictitious specific heat capacity is introduced for compensation as $\begin{array}{c} c_{e}^{*}=k_{m}^{-1}c_{e} \end{array}$ to balance the thermal time constant in the computation process [40].

The mass scaling method can significantly accelerate the computation speed of CEL models, but it will also reduce the accuracy of the computed results. To minimize the impact of inertial effects on the results, it is necessary to ensure that the ratio of kinetic energy to internal energy in the system is less than 10% [39]. Figure 7a shows a time comparison for the simulation experiments conducted using different mass scaling coefficients $k_{m}$. The analysis process involves a 12 s friction stir welding process, including a 2 s plunging stage and a 10 s welding stage. All parameters are the same except for the material density.

Figure 7a shows that as the mass scaling coefficient $k_{m}$ increased, the solution time stabilized at around 8.5 h. At the same time, the extracted forces data from the FEA model of the stirring tool are shown in Figure 7b–d. When $k_{m}=10^{5}$, the force curve was relatively stable with minor fluctuations. When $k_{m}=10^{6}$, the curve exhibited more significant fluctuations. When $k_{m}=10^{7}$, the force curve experienced a large fluctuation. To accelerate the computation speed while minimizing computational errors as much as possible, $k_{m}=10^{5}$ is chosen as the mass scaling coefficient.

#### 2.3.3. Co-Simulation Virtual Welding Model

The FEA model of friction stir welding can accurately replicate the changes in the temperature, stress, and material flow resulting from the action of the stirring tool during the friction stir welding process. It can also capture the reactive forces exerted on the stirring tool throughout this process. However, it falls short of reflecting the displacement of the stirring tool caused by the welding forces and the subsequent impact of this displacement on the welding quality. To enable the virtual welding model to replicate the interactions between the stirring tool and the welding material, a co-simulation model was established in this study, allowing the FEA model, the robot kinematic model, and the robot stiffness model to collaborate and exchange data. Through the co-simulation approach, these models were integrated to achieve a faithful representation of the robotic friction stir welding process, capturing the intricate interplay between the stirring tool and the welding material.

Due to the limitations of the finite element analysis process, which does not allow for real-time parameter changes, this study divides the welding process into smaller segments during the co-simulation. Suppose the welding force acting on the robot end-effector remains constant for each small segment. Then, the deviation between the current and expected positions was calculated based on the robot’s current pose. In the subsequent segment, the model will minimize the positional discrepancy to align the simulated outcomes with the actual welding results. The specific process is shown in Figure 8.

The co-simulation model started with inputting the actual welding parameters. The robot kinematic model transferred the initial position of the end-effector to the FEA model first. Then, the FEA model calculated the force on the stirring tool via the numerical computations and sent them to the robot stiffness model. The stiffness model computed the deflections of the robot joints and sent them back to the kinematic model. At this point, the model entered into the next iterative loop. Using the deflections of the joint, the kinematic model determined the end-effector’s deviation, calculated the robot end-effector’s displacements in the current segments, and then communicated these data to the FEA model. This iterative process continued until the virtual welding process was completed, ultimately resulting in the acquisition of forces and deviation data throughout the entire process.

This section has completed the establishment of the RFSW five-dimensional digital twin model and achieved the integration of the virtual welding model. In the Virtual Entity, the robot model can calculate the deviation in real time by reading the robot’s pose and end force in the Physical Entity. Moreover, before the actual welding, the virtual welding model can simulate and predict the welding results and the displacement of the robot end by reading data such as the robot’s teaching trajectory and welding parameters. However, the parameters of the virtual welding model are still based on theory and empirical values, resulting in a significant discrepancy from the actual welding results.

## 3. Calibration of Twin Data

Due to simplifications of the virtual welding model, the discrepancies between the theoretical and actual material parameters, and measurement errors from sensors during the actual welding process, the simulated data may exhibit disparities compared to the actual measurements. Comparing the twin data under identical welding parameters and calibrating simulated data with the actual measurements, optimizing the parameters of the virtual welding model can improve the model’s accuracy. The comparison of the forces data generated by the FEA model for the stirring tool and recorded by the sensor at the robot end-effector during the actual welding process is shown in Figure 9. The actual welding force was subjected to a sliding average filter with a window size of 30. The simulated force first underwent a median filter with a window size of 10 to filter out outliers, followed by a sliding average filter with a window size of 1000.

By comparing two sets of data under the same welding parameters, there are significant differences in the forces acting on the stirring tool. Figure 3c shows an analysis of the forces acting on the stirring tool during the friction stir welding process. The axial force ($F_{d}$) primarily resulted from the reaction force of the base material against the downward motion of the stirring tool, the resistance force ($F_{r}$) was mainly caused by the resistance of the base material to the forward motion of the stirring tool, and the lateral force ($F_{l}$) was primarily originated from the frictional forces between the stirring pin and shoulder and the base material during high-speed rotation. At this point, the base material is in a semi-molten state due to the heat generated by friction, making the contact forces with the stirring tool more complex. However, during the welding stage, the stirring tool and the base material are in a relatively stable state, allowing us to describe their tangential behavior using an equivalent friction coefficient, denoted as $f^{*}$.

Replace the friction coefficient in the FEA model with an equivalent friction coefficient $f^{*}$ and adjust its value. Record the simulated forces on the stirring tool for different equivalent friction coefficients and extract the average force on the stirring tool in the stable region of the force curve, as shown in Figure 9a. In the actual welding process, the average forces acting on the stirring head in a stable region are F_l0_ = 1350.87 N, F_d0_ = 4842.58 N, and F_r0_ = 899.80 N. Using a polynomial fitting method to fit the data in the table, it obtained curves representing the simulated forces in three directions as a function of the equivalent friction coefficient. As shown in Figure 9c, the points in the graph represent the average simulated forces of the stirring tool corresponding to different equivalent friction coefficients. At the same time, the curves were computed using a polynomial fitting method to represent the variations in the simulated forces.

It can be observed that as the equivalent friction coefficient increased, the absolute values of the welding forces gradually decreased. Among them, the variation in the axial force ($F_{d}$) is the most significant, while the lateral force ($F_{l}$) shows the smallest variation. The absolute value of the resistance force ($F_{r}$) decreased to below the lateral force when the equivalent friction coefficient is between 0.6 and 0.7. Considering the actual forces acting on the stirring tool during the welding process, the simulated forces are closest to the actual force data when the equivalent friction coefficient is between 0.9 and 1.0.

Establish a loss function based on the simulated force curve after fitting as Equation (15).
(15)$$\begin{array}{c} e={\left(F_{l}\left(x\right)-F_{l0}\right)}^{2}+{\left(F_{d}\left(x\right)-F_{d0}\right)}^{2}+{\left(F_{r}\left(x\right)-F_{r0}\right)}^{2} \end{array}$$
where $F_{l}\left(x\right)$, $F_{d}\left(x\right)$, and $F_{r}\left(x\right)$ are the functions of the welding force variation with respect to $f^{*}$ after fitting.

e represents the comprehensive error between the simulated force on the stirring tool corresponding to the current x value and the actual force. When the loss function reached its minimum value, the comprehensive error between the simulated force and actual force was minimized. At this point, the equivalent friction coefficient was $f^{*}=0.935$. Add the calculated equivalent friction coefficient to the virtual welding model to complete the calibration of the twin data.

## 4. Analysis of the Virtual Welding Results

Input the welding parameters and the starting and ending position of the robot end-effector (as shown in Table 6) into the virtual welding model. The material properties of the base material and the stirring tool were replicated using the same materials as in actual welding processes, specifically AL6061-T6 aluminum alloy for the base metal and H13 tool steel for the stirring tool. In the virtual welding process, experiments were conducted employing three plunging distances: 7 mm, 8 mm, and 9 mm, respectively. The plunging distance refers to the distance continued insertion commanded after the robot end-effector makes contact with the surface of the base material rather than the actual plunging depth of the stirring tool into the base material. The motion is defined within a Cartesian coordinate system, where the robot end-effector stirring tool is directed to plunge in the negative direction of the *z*-axis and feed along the negative direction of the *y*-axis for the welding stage. The virtual welding model generated the final simulated weld seam results shown in Figure 10.

Figure 10a–c illustrate the displacement of the robot end-effector stirring tool during the plunging and dwelling stages, corresponding to three distinct plunging distances. The green curve represents the actual displacement of the robot end-effector stirring tool as it plunges the base material. The blue and red curves denote the robot end-effector’s lateral deviations along the *x*- and *y*-axis. The yellow dashed line indicates the thickness of the base material. The graph shows that at approximately 0.5 s, the stirring tool makes contact with the surface of the base material, followed by a continued downward movement. Subsequently, the plunging commands are completed around 4 s, 4.5 s, and 5 s, respectively. After which, the process enters a dwelling stage. Due to insufficient heat input to the base material, it has not been adequately softened. Under the applied downward force, the stirring tool continues to plunge further. At approximately 6 s, the stirring tool reaches its maximum in both the *X* and *Y* directions, signifying the initiation of the base material’s softening. Figure 10a shows that when the plunging distance is set to 7 mm, the penetration depth of the stirring tool stabilizes at approximately 4.2 mm around 16 s. However, due to insufficient downward pressure, this depth does not meet the requirements for welding. In Figure 10b, the horizontal position of the stirring tool enters a relatively stable state around 12 s, indicating that the base material has sufficiently softened. The penetration depth increases gradually, stabilizing at around 4.8 mm at around 16 s, which meets the welding requirements. Conversely, in Figure 10c, the stirring tool achieves a relatively stable state around 15 s. However, by around 12 s, the depth has already exceeded the thickness of the base material, resulting in excessive penetration when stability is reached.

Subsequently, the welding stage is activated when the end-effector stirring tool achieves a stable position. Figure 10d–f depict the virtual welding results corresponding to three different plunging distances, while Figure 10g–i present cross-sectional views of the weld seams. In all outcomes, the weld trajectories exhibit a lateral deviation of 5–7 mm, consistent with the results in actual welding processes. Figure 10d shows that defects are observed on the surface and within the weld seam due to an insufficient plunging distance, as illustrated in Figure 10g. However, in Figure 10f, excessive penetration results in the extrusion of the base material, leading to accumulation on the left side of the seam in the feed direction and excessive thinning of the base material, as depicted in Figure 10i. Inappropriate welding parameters not only lead to a decline in the surface quality of the weld seam but also affect the evolution of the microstructure of the metal. Compared to unsuitable welding parameters, the average grain size in the welding area is more refined under appropriate parameters, resulting in a better tensile strength and lower residual stress at the joint [8,41]. In contrast, Figure 10e demonstrates an appropriate level of penetration depth, where the weld seam region exhibits no significant thinning or defects. Furthermore, in the final ten iterations of the cycle calculation, the average lateral deviation in the weld seam in the virtual welding process was 5.24 mm. In the actual welding process with the same parameters, as shown in Figure 3b, the actual displacement of the weld seam was approximately 4.9 mm. Compared to the actual welding, the deviation in the weld seam lateral offset obtained through virtual welding was about 6.9% of the actual value. By analyzing the residual stress field in the results, it can be found that when plunging 8 mm, the residual stress near the weld is relatively uniform, with the maximum appearing on the advancing side (AS) surface near the weld’s start. When plunging 7 mm, more considerable residual stress occurs near the internal defects. Meanwhile, when plunging 9 mm, more considerable residual stresses appear on the AS surface near the weld’s start and internally, with a maximum value greater than plunging 8 mm. This indicates inappropriate plunging distances can cause excessively large and uneven residual stress inside the weld.

From the results of the virtual welding experiments above, it can be deduced that the digital twin virtual welding model can reflect the outcomes of robotic friction stir welding under varying parameters. The results of the virtual welding are consistent with those from actual welding processes.

Based on digital twin technology, the virtual welding model exhibits good applicability and expandability. Due to the co-simulation approach, the robot model and FEA model are independent. They operate in different programs, exchanging data through script files. Only parameter adjustments or model replacements are needed when matching other robot models and welding tasks. However, this study still has certain limitations, particularly regarding simplifications in the model and errors from experimental methods. These include errors from the simplification of the robot model, errors from the stiffness identification experimental methods, and discrepancies between the finite element model’s attributes and actual parameters, among others. By developing more precise models, employing more reliable experimental methods, and measuring parameters that better reflect reality, a more accurate RFSW digital twin model can be established.

## 5. Conclusions

This study focuses on the challenge of measuring end-effector deviation in RFSW due to the insufficient stiffness of serial industrial robots. A digital twin model for RFSW was established using the five-dimensional digital twin theory, and a virtual welding model was developed through co-simulation, enabling the welding process simulation and end-effector deviation prediction. The conclusions of this study are as follows:The RFSW digital twin model effectively reflects the working status of the robot, achieving real-time monitoring of the robot end-effector displacement.Analysis of the virtual welding experiments with three different plunging depths shows that the virtual welding model can accurately reflect the interaction between “force-heat-displacement” during welding.Comparing the results of virtual welding with actual welding under the same parameters indicates that the virtual welding model can predict the robot end-effector deviations. The calculated results show an approximate error of 6.9% compared to the actual values.The RFSW digital twin and the virtual welding model still have limitations. Improved accuracy in prediction can be achieved through model refinement and method optimization.

Digital twin technology digitizes and informatizes traditional manufacturing, enabling integration with cloud computing, big data, and artificial intelligence. The RFSW digital twin model can also be combined with intelligent algorithms to achieve automatic compensation for welding errors, use cloud services for remote monitoring of the manufacturing process, and utilize deep learning to optimize welding parameters.

## Figures and Tables

**Figure 1 sensors-24-01001-f001:**
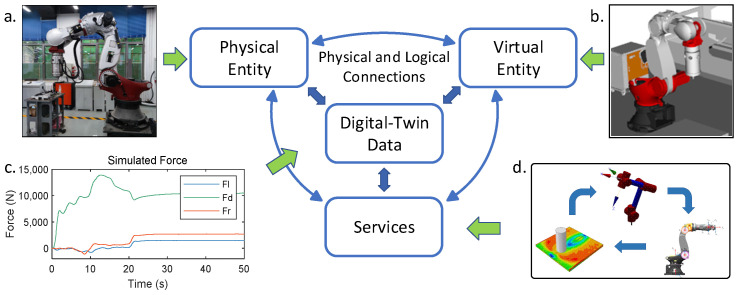
Five-dimensional digital twin model of robotic friction stir welding. (**a**) FSW robot and working conditions. (**b**) Simulation model in the Virtual Entity. (**c**) Simulated force generated by virtual welding model. (**d**) Virtual welding model in the Services module. The blue arrows indicate physical and logical connections, while the green arrows represent the correspondence between module and their contents.

**Figure 2 sensors-24-01001-f002:**
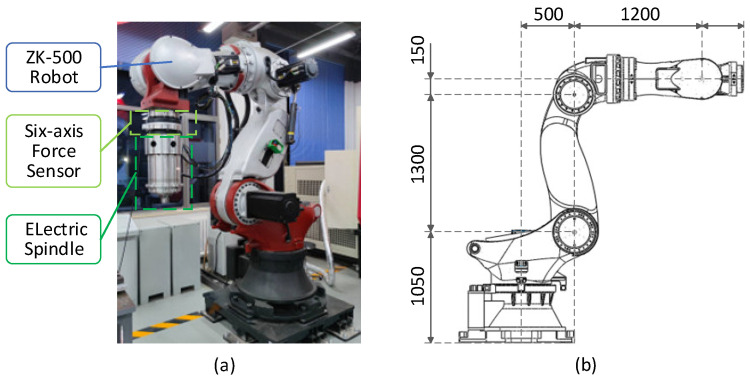
Friction stir welding robot. (**a**) The composition of the friction stir welding robot. (**b**) Dimensional parameters of the ZK-500 robot.

**Figure 3 sensors-24-01001-f003:**
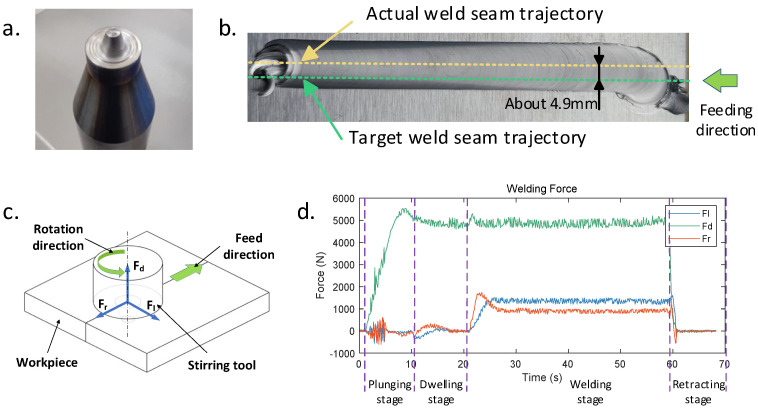
Results of robotic friction stir welding. (**a**) Conical stirring tool. (**b**) Weld seam with a displacement of about 4.9 mm. (**c**) Force analysis of the stirring tool during welding. (**d**) Welding force recorded by the six-axis force sensor.

**Figure 4 sensors-24-01001-f004:**
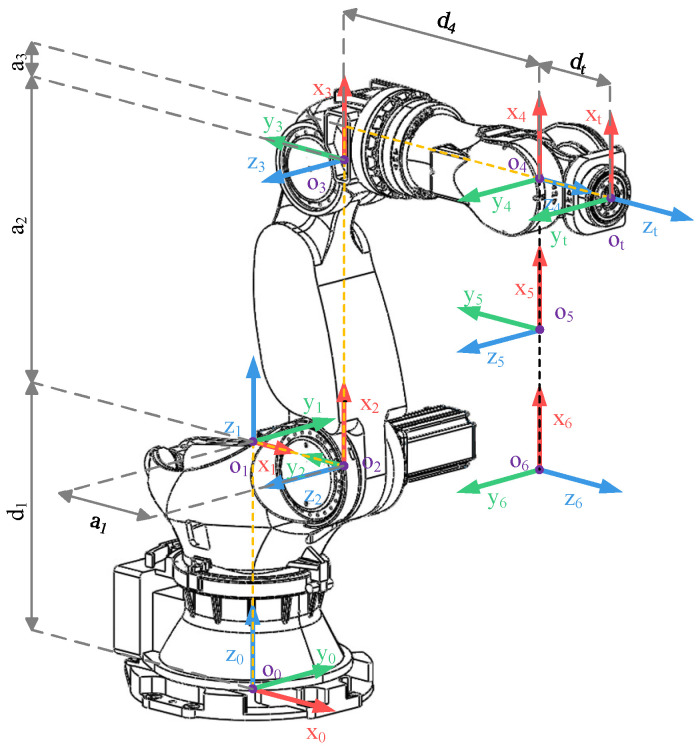
MDH model of ZK-500.

**Figure 5 sensors-24-01001-f005:**
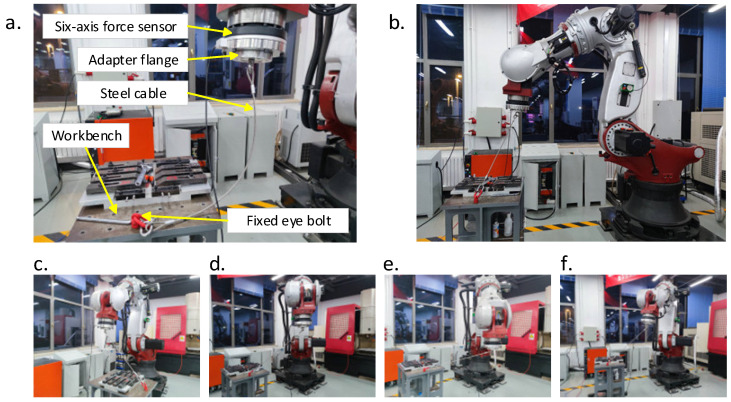
Robot joint stiffness identification experiment. (**a**) Equipment for the stiffness identification experiment. (**b**–**f**) Different poses of the robot.

**Figure 6 sensors-24-01001-f006:**
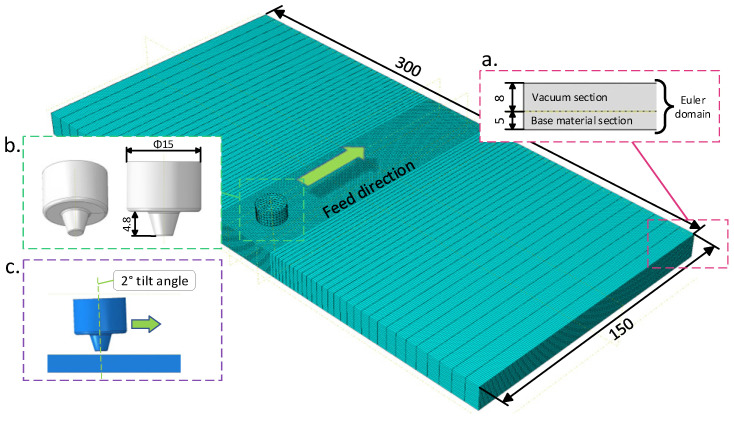
Assembly of the weldment and stirring tool model. (**a**) Sections of the Euler domain. (**b**) Dimensions of the stirring tool model. (**c**) The tilt angle of the stirring tool along the feed direction.

**Figure 7 sensors-24-01001-f007:**
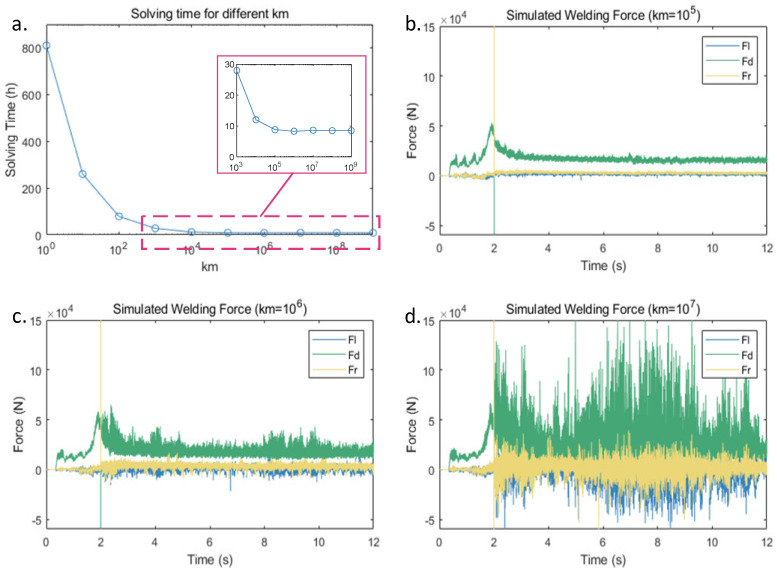
Comparison of the results for different mass scaling coefficients. (**a**) Solving time variation curve for different mass scaling coefficients. (**b**) Simulated welding force when $k_{m}=10^{5}$. (**c**) Simulated welding force when $k_{m}=10^{6}$. (**d**) Simulated welding force when $k_{m}=10^{7}$.

**Figure 8 sensors-24-01001-f008:**
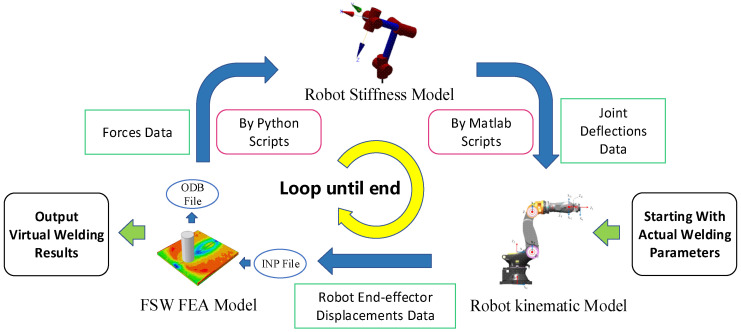
Co-simulation model of robotic friction stir welding.

**Figure 9 sensors-24-01001-f009:**
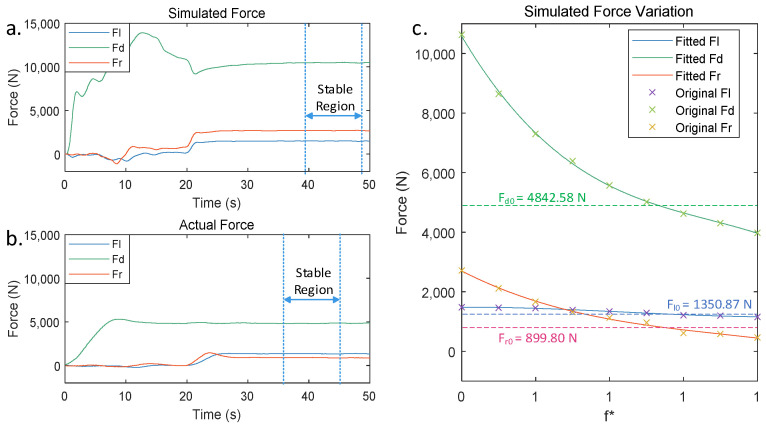
Comparison of simulated force in virtual welding and recorded force in actual welding of FSW. (**a**) Simulated force of virtual welding process. (**b**) The real force of the actual welding process. (**c**) Simulated force variation in different equivalent friction coefficients.

**Figure 10 sensors-24-01001-f010:**
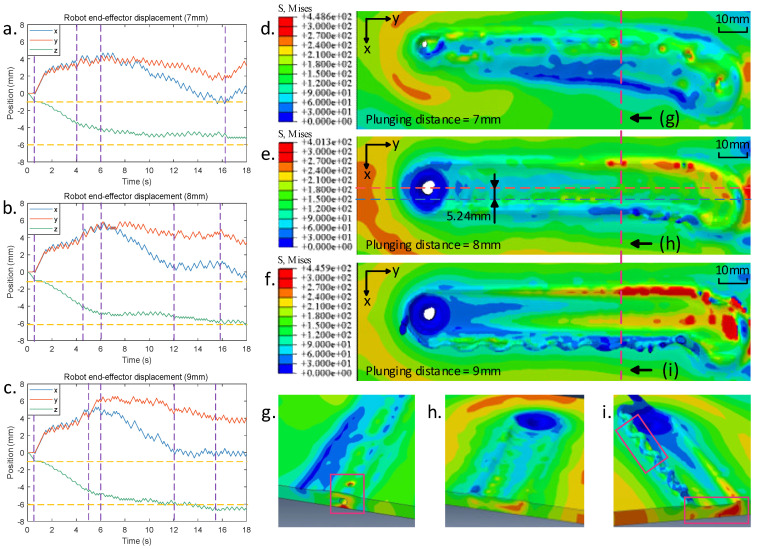
Comparison of virtual welding results of different plunging distances. (**a**–**c**) The displacement curves of the robot end-effector during the plunging and dwelling stages when the plunging distances are 7, 8, 9 mm, respectively. (**d**–**f**) The von Mises stress field in the complete virtual welding results when the plunging distances are 7, 8, 9 mm, respectively. (**g**–**i**) The internal stress state and defects in the virtual welding seam when the plunging distances are 7, 8, 9 mm, respectively.

**Table 1 sensors-24-01001-t001:** MDH parameters of ZK-500.

i	$\theta_{i}$ (°)	$d_{i}$ (mm)	$a_{i-1}$ (mm)	$\alpha_{i-1}$ (°)
1	$\theta_{1}$	1050	0	0
2	$\theta_{2}$ + 90	0	500	90
3	$\theta_{3}$	0	1300	0
4	$\theta_{4}$	1200	150	90
5	$\theta_{5}$	0	0	−90
6	$\theta_{6}$	0	0	90
t	0	390	0	0

**Table 2 sensors-24-01001-t002:** Joint angles of different poses.

Pose	$\theta_{1}$ (°)	$\theta_{2}$ (°)	$\theta_{3}$ (°)	$\theta_{4}$ (°)	$\theta_{5}$ (°)	$\theta_{6}$ (°)
1	−96.24	3.91	25.45	−2.10	57.17	2.90
2	−96.10	23.70	−1.25	1.48	66.89	1.50
3	−49.90	13.09	13.03	1.48	63.22	4.94
4	−43.38	18.77	27.96	1.90	42.54	7.15
5	−53.40	−5.53	30.57	−3.90	59.00	3.19

**Table 3 sensors-24-01001-t003:** Stiffness values of each joint.

Joint	Stiffness (Nm/rad)
1	1.1676 × 10^6^
2	1.2731 × 10^8^
3	1.7187 × 10^6^
4	7.3227 × 10^5^
5	1.1011 × 10^6^
6	8.8932 × 10^6^

**Table 4 sensors-24-01001-t004:** Temperature-dependent material properties for AA6061-T6.

Temperature (°C)	Density (kg/m^3^)	Young’s Modulus (MPa)	Yield Strength (MPa)	Poisson’s Ratio	Thermal Expansion (μm/m·K)	Thermal Conductivity (W/m·K)	Specific Heat (J/kg·K)
25	2700	68,900	276	0.33	22	167	896
37.8	2685	68,540	274.4	0.331	23.45	170	920
93.3	2685	66,190	264.6	0.334	24.61	177	978
148.9	2667	63,090	248.2	0.335	25.67	184	1004
204.4	2657	59,160	218.6	0.336	26.6	192	1028
260	2657	53,990	159.7	0.338	27.56	201	1052
315.5	2630	47,480	66.2	0.36	28.53	207	1078
371.1	2620	40,340	34.5	0.4	29.57	217	1104
426.7	2602	31,720	17.9	0.41	30.71	223	1133

**Table 5 sensors-24-01001-t005:** Material properties for H13.

Density (kg/m^3^)	Young’s Modulus (MPa)	Poisson’s Ratio	Specific Heat (J/kg·K)
7800	210,000	0.3	460

**Table 6 sensors-24-01001-t006:** Virtual welding parameters.

Parameter Name	Value
Rotation speed (rpm)	1500
Welding speed (mm/s)	3
Plunging distance (mm)	7, 8, 9
Plunging speed (mm/s)	2
Starting coordinate (mm)	(269.52, −2070.91, 663.15)
Ending coordinate (mm)	(278.10, −1957.60, 664.05)
Iteration interval (s)	0.1

## Data Availability

Data are contained within the article.

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
