# Peer review of "Digital Twin Virtual Welding Approach of Robotic Friction Stir Welding Based on Co-Simulation of FEA Model and Robotic Model"

_sensors, 2024, doi:10.3390/s24031001_

Round 1
Reviewer 1 Report
Comments and Suggestions for Authors
The work provides a persuasive analysis of how to overcome the difficulties caused by the inherent low rigidity of serial industrial robots in the context of robotic friction stir welding. Utilising a digital twin model that includes a six-axis force sensor, robot kinematic, and stiffness models, offers a unique method to accurately compute real-time end-effector deviations. By incorporating a virtual welding model and utilising finite element analysis of friction stir welding, the study's breadth is expanded. This enables the modelling and optimisation of welding parameters before the actual welding process takes place. The congruence between virtual welding outcomes and real-world results demonstrates the efficacy of the proposed methodology in simulating and forecasting actual welding situations.
· Keywords should be improved. They should not be the same as the title of the paper. It will help in improving the sustainability.
· The paper efficiently connects the theoretical models (FEA) and practical implementation (robotic model), providing a comprehensive perspective on the suggested digital twin virtual welding approach. The incorporation of this integration is crucial for the effective implementation of such techniques in practical situations.Nevertheless, in order to enhance the paper's robustness, the authors should explore more extensively the particular obstacles and constraints related to the incorporation of 3D printing within the framework of their study. This would offer readers a more intricate comprehension and maybe create opportunities for further investigation.Authors may include these studies if found appropriate: https://ieeexplore.ieee.org/abstract/document/10337499; https://www.taylorfrancis.com/chapters/edit/10.1201/9781003306238-5/identification-overcoming-key-challenges-3d-printing-revolution-ashish-kaushik-upender-punia-sumit-gahletia-ramesh-kumar-garg-deepak-chhabra;
· The first section of the introduction may be linked to global causes like SDGs or net-zero targets.
· How did you decide on the different parameters mentioned in Table 1.
· Please provide further information pertaining to the experimental configuration employed for validating the proposed digital twin model. Readers would benefit from having information regarding the robotic system, the six-axis force sensor, and the unique welding conditions in order to evaluate the suitability and universality of the approach.
· What methods were used to verify the accuracy of the virtual welding model in comparison to real welding results? Are there designated trials or scenarios employed to showcase the precision and dependability of the virtual predictions?
· To what extent can the proposed digital twin paradigm be used to various robotic platforms utilised in friction stir welding? Are there any specific factors or adjustments required for its implementation in various industrial environments?
· What difficulties were faced when implementing the proposed model in a real-world robotic friction stir welding environment, considering the focus on measuring end-effector deviation in real-time? How does the model adapt to dynamic changes that occur during the welding process?
· Used equations should be cited.
· How the authors handled the uncertainty in the measurement of data.
· Explore avenues for future research in this area. Suggest potential experiments, modeling improvements, or practical applications that can build upon the current study and expand our knowledge.
· Take care of typos and grammar
Pay special attention to subscripts and superscripts.
Overall, it is a well-written and compact study. It may be considered after revising,

Fine, minor changes required.
Author Response
Dear Reviewer,
First and foremost, I would like to express my sincere gratitude for the time and effort you have dedicated to reviewing our manuscript and providing valuable feedback. We have carefully considered each of the comments and have made thorough revisions to our paper to better reflect these insightful suggestions. The detailed response please see the attachment.
I appreciate the opportunity to enhance my work through your insights.
Best regards,
Guanchen Zong

Reviewer 2 Report
Comments and Suggestions for Authors
This work could be a milestone in the field of the joining process. However, it requires adequate explanation on several questions or comments below for the review. The authors are encouraged to explain and answer these:
1- English should be improved in all the papers.
2- A concise and factual abstract is required. It should contain the objective, methods, results, and conclusions, with emphasis on the results and conclusions.
3- It should clearly state the key novelty point of your work.
4- The introduction section is written very complicated. It is too long and wordy, too.
5- Actually, in the introduction, the framing of the sentences is not acceptable. Regarding the advances in the FSW, much has been done to improve the process by using accessory equipment such as friction stir vibration welding (FSVW). As you know, the primary aim of the study in the introduction section is to provide a summary report from the latest investigations of the FSW method, especially FSVW.
6- The last paragraph of the introduction section must summarize the investigation of the current work with its novelty compared to previous studies. The objective of the work and the method used during the investigation. The current format is too complicated and wordy.
7- The section “Digital twin virtual welding model of robotic friction stir welding” must be summarized.
8- Based on what critical the welding parameters were selected?
9- The quality of nearly all Figs is poor and is not acceptable for publication.
10- While experimental results showed improvement in processing quality, a deeper explanation is lacking in some sections. For example, the recrystallization process and dislocation density during the FSW will play a prominent role in microstructure evolution and mechanical properties. The author must use the following literature to cover this point:
- https://doi.org/10.1177/14644207211044407
- https://doi.org/10.1007/s12289-021-01632-w
11- More experimental analysis should be presented in the revised version. For example, OM analysis of microstructure evolution would be useful.
12- A concise and factual conclusion is required. It is too general.
Comments on the Quality of English LanguageMinor Revision is needed.
Author Response

(The authors gave the same response as above.)

Reviewer 3 Report
Comments and Suggestions for Authors
The manuscript titled “Digital Twin Virtual Welding Approach of Robotic Friction Stir Welding Based on Co-simulation of FEA Model and Robotic Model” discusses an important in robotic FSW and the use of digital twin technology. However all section needs significant improvements in terms of coherency and consciences
The introduction section needs significant improvement in terms of coherency and consciences, and also limited literature was covered on digital twin in friction stir welding or even FSW in general. Some suggested references in the mentioned topic might be used to enrich both topics.
10.3390/met10070914; 10.3390/ma16082971; 10.1016/j.jmapro.2021.01.042
10.1016/j.jma.2023.09.039; 10.3103/S1068798X21040146; 10.3390/cryst13111559
A clear and concise objective needs to be given at the end of the introduction section instead of just summarizing the manuscript's content.
Figure 1 needs rearrangement and improvment ; suggest starting with the physical entity module and then the other modules. The text begins first by talking about the physical entity.
The virtual welding results presented in Figure 10 indicate a deflection in the welding direction after the plunging stage, is there any explanation for that, and does that occur in the actual welding process?
Is the virtual welding module able to give feedback to the physical welding entity in order to corret the end effector displacement indicated I the work?
Comments on the Quality of English LanguageIt looks ok
Author Response

(The authors gave the same response as above.)

Reviewer 4 Report
Comments and Suggestions for Authors
The objective of this paper is to introduce a digital twin model for robotic friction stir welding, combining force sensor with robot kinematic and stiffness models. It is integrated with a finite element analysis model which enables pre-process simulation resulting in outcome prediction, deviation compensation, and parameter optimization. The application seems to be very useful; some remarks and comments are added in order to upgrade quality of the paper.
In Lines 391-392, effective plastic strain rate is denoted twice, please correct it.
In Lines 394-395, please correct the units of constants of the Johnson–Cook constitutive model, such as use MPa instead of Mpa, and use spaces after each colon.
Please show the finite element mesh and demonstrate what is the minimum element size for the calculation of time increment in the explicit analysis.
Please name the software used for the finite element analysis and add a reference with version number included.
Please add the description of element type used in the analysis.
In Lines 487-488, please clarify the applied type of filtering and smoothing algorithms.
Please add a table or figures about both temperature-dependent thermal and mechanical material properties used in the FE model.
Is there any initial contact modelled between the two plates to be welded?
Author Response

(The authors gave the same response as above.)

Round 2
Reviewer 1 Report
Comments and Suggestions for Authors
The revised paper can be accepted for publication.
Reviewer 2 Report
Comments and Suggestions for Authors
Accept
Reviewer 3 Report
Comments and Suggestions for Authors
All comments were considered by the authors